# Household Food Security and Consumption of Sugar-Sweetened Beverages among New York City (NYC) Children: A Cross-Sectional Analysis of 2017 NYC Kids’ Data

**DOI:** 10.3390/nu15183945

**Published:** 2023-09-12

**Authors:** Karen R. Flórez, Sandra S. Albrecht, Neil Hwang, Earle Chambers, Yan Li, Francesca M. Gany, Marivel Davila

**Affiliations:** 1Environmental, Occupational and Geospatial Sciences Department, Graduate School of Public Health and Heath Policy, City University of New York, New York, NY 10017, USA; 2Department of Epidemiology, Mailman School of Public Health, Columbia University, New York, NY 10032, USA; ssa2018@cumc.columbia.edu; 3Business and Information Systems Department, Bronx Community College, City University of New York, Bronx, NY 10453, USA; neil.hwang@bcc.cuny.edu; 4Department of Family and Social Medicine, Albert Einstein College of Medicine, Bronx, NY 10461, USA; earle.chambers@einsteinmed.edu; 5Department of Population Health Science and Policy, Icahn School of Medicine at Mount Sinai, New York, NY 10029, USA; yan.li1@mountsinai.org; 6Immigrant Health and Cancer Disparities Service, Memorial Sloan Kettering Cancer Center, New York, NY 10065, USA; ganyf@mskcc.org; 7Bureau of Health Promotion for Justice-Impacted Populations, New York City Department of Health and Mental Hygiene, New York, NY 11101, USA; marivel.davila@gmail.com

**Keywords:** NYC kids’ data, food security, social inequities, sugar-sweetened beverages, sugary drinks, Latinos, immigrants

## Abstract

Food insecurity is a stressor associated with adverse health outcomes, including the consumption of sugar-sweetened beverages (SSBs). Our study tests the hypothesis that other socioeconomic vulnerabilities may magnify this effect using cross-sectional data from the 2017 New York City (NYC) Kids Survey. Households providing an affirmative response to one or both food security screener questions developed by the US Department of Agriculture were coded as households with low food security. The number of sodas plus other SSBs consumed was standardized per day and categorized as 1 = none, 2 = less than one, and 3 = one or more. We tested the joint effect of low food security with chronic hardship, receipt of federal aid, and immigrant head of household on a sample of *n* = 2362 kids attending kindergarten and beyond using ordinal logistic regression and accounting for the complex survey design. Only having a US-born parent substantially magnified the effect of low household food security on SSB consumption (OR = 4.2, 95%CI: 2.9–6.3, *p* < 0.001) compared to the reference group of high household food security with an immigrant parent. The effect of low food security on SSB consumption among NYC children warrants intersectional approaches, especially to elucidate US-based SSB norms in low-food-security settings.

## 1. Introduction

Low levels of household food security, defined as a lack of consistent access to enough food to support an active healthy life, are associated with poverty and negative outcomes related to health, educational attainment, and social mobility [1,2,3,4,5,6,7]. Low food security is also associated with the presence of children: 12.5% of US households with children experienced low food security, compared to 10.2% of US households overall, in 2021 [8].

Research is increasingly focusing on the negative dietary consequences of low household food security levels, and findings suggest that households with fewer monetary resources tend to have less healthier eating habits [9]. For example, studies looking at households with low food security suggest that people in them are more likely to think of energy-dense items like SSBs as “cheap” and “better value than water” [10], and taste is a major reason for consumption among children and young adults [11]. Qualitative research also highlights that parents perceive sugary treats, including soda, as small pleasures that they should grant to their children and that they have a powerful symbolic value for families that have to sacrifice on a regular basis [12]. Research on patterns also suggests the importance of parents in determining the SSB intake of their children as well as the time and location of this intake. For example, studies have shown that low food security is significantly associated with greater SSB intake among children when they are out of school, such as in the summer [13]. This relationship was consistent for younger children [14,15], adolescents [16], and young adults [17,18]. Children with high SSB consumption typically consume SSBs at home, including during meals and in the evenings [19].

The data from the CDC’s Youth Risk Behavior Surveys (YRBSs) corroborate a higher prevalence of sugar-sweetened beverage (SSB) consumption among youth in lower-socioeconomic-status (SES) households [20]. However, the national YRBS does not have a measure of household food security, has limited sociodemographic indicators (e.g., race/ethnicity, free and reduced-price lunch eligibility), and only samples students in grades 9–12 [21]. However, a few studies have focused on socioeconomic factors that may magnify or offset the negative dietary consequences of low food security among children. For example, some evidence suggests that federally funded programs designed to address food security, a variable examined here, have differential effects on SSB consumption among children. In one study using multiple waves of data from the National Health and Nutrition Examination Survey, Supplemental Nutrition Assistance Program (SNAP) benefits were associated with greater SSB consumption, but the benefits from the Special Supplemental Nutrition Program for Women, Infants, and Children (WIC) were not [22]. Parental nativity as a proxy for immigration, another variable examined here, may also be important, for multiple reasons. On the one hand, immigrant families may have more difficulty accessing SNAP, WIC, and other food insecurity amelioration programs [23]. On the other hand, the dietary acculturation process immigrant children and their families face is replete with complexities. Concerns about the impact of acculturation on their children may exacerbate the dynamic in which low-income parents provide sugary treats to their children to offset the challenges of poverty [14]. Epidemiological research suggests that greater consumption of processed foods in general is associated with greater acculturation to the US, and the same dynamic may apply to SSBs [24]. Animal studies and psychological experiments have also shown an important link between glucose consumption as a coping strategy under stressful circumstances, which is a possible reason why low food security may heighten SSB consumption and why chronic stress is examined here specifically. No study of food security and SBB consumption focuses on New York City, the context for the current study. The one such study that focuses on New York State adults [20] found that SSB consumption remained high among African Americans and Latinos in 2016 despite decreases across time.

This paper investigates whether and, if so, how the receipt of federal aid, parental immigration status, and chronic stress modify the association between low food security of households and SSB consumption among children. We draw on data from the Healthy People 2030 framework of Social Determinants of Health (SDOH) [25] to examine how other important socioeconomic factors may serve to modify the effect of household food security levels on SSB consumption in children (see Figure 1). We hypothesize that the three variables of interest would magnify the main effect among children experiencing low levels of household food security, causing them to have a higher SSB consumption (main effect). New York City is an excellent setting to test this hypothesis, given that low food security was as high as 12.6% before 2019, but few datasets exist to derive estimates at this level of geographic granularity. This is valuable since we know that food security rates are not uniform across the US, and more research is needed to understand the micro-level policies and economic conditions that give rise to these differences in prevalence [8]. For example, the average meal cost in NYC is double the national average (e.g., USD 6.31 vs. USD 3.25), and the high cost of food may be particularly impactful among at-risk families [22].

## 2. Materials and Methods

### 2.1. Data Source

The NYC Department of Health and Mental Hygiene conducted the NYC KIDS Survey in 2017, which provides estimates that are representative of NYC children ages 0–13. The sampling frame included 7507 households with one or more child/children aged 0–13 years across all five NYC boroughs: Bronx, Brooklyn, Manhattan, Queens, and Staten Island. The survey data were collected via telephone with parents, guardians, or other family members knowledgeable about the randomly selected child’s health, doctor visits, and family and neighborhood characteristics, conducted in their preferred language if other than English. The questions were drawn from national surveys on children’s health, such as the National Health Interview Survey and the National Survey of Children’s Health, as well as prior NYC surveys, such as the 2009 NYC Child Community Health Survey and the 2015 NYC Child Health, Emotional Wellness, and Development Survey. A data use agreement between the authors and the NYC Department of Health and Mental Hygiene was obtained for the present study. The Institutional Review Board of the NYC Department of Health and Mental Hygiene determined that the activity was exempt (Re:20-068).

### 2.2. Outcome Variable

Overall sweetened beverage consumption per day: The data contained a constructed variable based on the number of sodas and other SSBs consumed, standardized per day, and categorized as follows in regression models: 1 = none, 2 = less than one, 3 = one or more. For soda, the parents/guardians were asked to report how often their child drank sugar-sweetened soda per day. For other SSBs, they were asked to report how often their child consumed sweetened drinks, such as sweetened iced tea, sports drinks, fruit punches, or other fruit-flavored drinks, excluding diet soda, sugar-free drinks, or 100% juice. Unfortunately, there were no specific probes for culturally specific SSBs, such as aguas frescas, which Latino parents perceive as healthy due to their “natural” ingredients [23].

### 2.3. Exposures

Household food security: A validated 2-item screener from the US Department of Agriculture (USDA) was used to identify families’ levels of household food security [8]. The parent/adult caregivers of the NYC Kids Survey responded on a 5-point Likert scale (very often, often, sometimes, rarely, or never) to the following items: (1) “Within the past 12 months, we worried whether our food would run out before we got money to buy more” and (2) “Within the past 12 months, the food we bought just didn’t last and we didn’t have money to get more.” Following the USDA’s standards for scoring, households were classified as having low food security if their responses were affirmative (very often, often, or sometimes) to one or both of the items.

Chronic stress: Respondents were asked to rate the extent to which it was difficult “to cover the basics (e.g., housing) since the child was born” on a 4-point Likert scale that ranged from very often, somewhat often, not very often, never. Affirmative responses were coded as follows: 1 = chronic hardship (very often, somewhat often) and 0 = no chronic hardship (not very often, never). Given the similar nature of chronic hardship and low food security, we tested for collinearity using a variance inflation factor (VIF). Our result was 1.34, far below the VIF value of 10 that may merit further investigation [24].

Receipt of federal aid: The parents/guardians were asked, “At any time in the past 12 months, even for one month, did you or anyone living or staying with you receive any cash aid from the Family Assistance program or Temporary Aid to Needy Families Program (TANF/TAN-IF), Food Stamps (referring to SNAP) or Electronic Benefit Transfer (EBT), or any other benefit or welfare programs?”.

Parental immigration status: The respondents were asked about their place of birth; their responses were coded as 1 = US if the parent was born in the continental US and any of the US territories and 0 = immigrant for anyone born outside of the US/US territories.

Confounders: At the child level, we used age (continuous), sex (female, male), and race and ethnicity (non-Latino White, non-Latino Black, non-Latino Asian, Latino, non-Latino Other). At the household/parental level, these variables included the age of the adult respondent (16–24; 25–44; 45–65+), their highest educational level (elementary school; 9–11; high school or GED (General Equivalency Diploma); college 1–3 years; college 4+), children in households other than those sampled (none; one; two; three; three or more), and whether the household was above or below the federal poverty level (FPL; <200%, 200%+).

### 2.4. Analysis

As children age, they have greater opportunities for unhealthy beverage consumption through socialization in school [26]. Hence, we only included children attending kindergarten or beyond. We excluded children who were too young to go to school or those attending pre-kindergarten (pre-K) (*n* = 4522) as well as children who were homeschooled (*n* = 29) or for whom the type of school they attended was unknown (*n* = 4). We then listwise deleted observations with missing data on outcomes (*n* = 46 soda; *n* = 54 for any other SSBs), food security (*n* = 71), and any covariates (*n* = 419) for a complete-case analytical sample size of *n* = 2362 school-aged children.

About 13.1% of the observations in the original dataset had missing values in at least one variable intended for our analysis. The percentage of missing values by variable ranged from 2.7% for highest parental education to less than 0.01% for parent’s age. We assessed the patterns of missingness for all the variables to be used in our analysis and found that patterns in the missing data were largely consistent with the missing completely at random (MCAR) assumption, and we therefore conducted the listwise deletion of missing values and performed the analysis on complete cases. 

The initial analyses included bivariate analysis using the chi-square test for our primary predictor variable, household food security, and our outcome variables of SSB consumption with related covariates. Given the multinomial ordinal response variable, we used ordinal logistic regression to estimate odds ratios (ORs) and their corresponding 95% confidence intervals. Ordinal logistic regression is an extension of logistic regression that is suitable for analyzing ordinal responses. We hypothesize a specific ordinal regression model called the proportional odds model (also known as the cumulative logic model) [27,28], which takes the following form: logPY≤yi|xjPY>yi|xj=αi−xj′β
where yi is an ordinal outcome i and xj are participant j’s values for some k covariates with αi∈ℝ and β∈ℝk. While this model form allows each ordinal level a different intercept αi, it assumes an identical log-odds ratio β for all outcome levels, which is known as the parallel lines assumption. This contrasts with alternative model forms such as the partial proportional odds model that allows different log-odds, i.e., βi, to be estimated for each outcome level. We assessed the appropriateness of the parallel lines assumption by performing the Brandt test and found that the parallel regression assumption in the proportional odds model holds for each model we discuss in this paper. 

Consistent with previous studies testing the joint effect of food insecurity [29], we examined the effects of low food security with chronic stress, receipt of federal aid, and parental immigration with SSB. Specifically, we tested the joint effect of low food security and chronic stress by classifying children into (a) high food security/no stress (reference), (b) high food security/stress, (c) low food security/no stress, or (d) low foods security/stress. This same 4-level categorization scheme was followed for the other joint effect models, using as the reference category the stratum with the lowest risk following best practices [30]. All marginal effects were computed for each joint effect model, except for immigration status since this variable was hypothesized to be an effect modifier. We also used Stata survey commands, which enabled us to adjust for the survey weighting [31]. The statistical significance was set at *p* < 0.05, and all tests were two-tailed.

## 3. Results

Descriptive characteristics at the child and household/parent levels, by food security level, are shown in Table 1. The food security level did not predict the children’s age or sex. However, households with low food security had a disproportionately higher percentage of Latino (51.4%) and Black non-Latino children (26.9%). In terms of household- and parent-level characteristics, households with low food security had disproportionately higher percentages of parents aged 25–44 and were categorized below the federal poverty level. 

They also had a higher prevalence of chronic stress, receipt of aid, being headed by an immigrant parent, and having parents/guardians with a high school or lower level of educational attainment. Figure 2 outlines the SSB consumption for children as reported by a parent or guardian based on household food security levels. A higher percentage of children in low-food-security households consumed one or more SSBs (43%) compared to children residing in high-food-security households (26%). Conversely, a lower percentage of children in low-food-security households consumed no SSBs (17%) compared to children in high-food-security households (30%). Table 2 shows both the crude and adjusted associations between household food security and SSB consumption. After adjusting for confounders, we found that children living in households with low food security had higher odds of consuming SSBs (OR = 2.00, 95% CI:1.6–2.6, *p* < 0.001).

Table 3 presents the summary of regression models to investigate whether chronic stress modifies the effect of household food security on its children’s SSB consumption. It shows that, within the no chronic-stress stratum, chronic stress magnifies the effect of low food security, while its effect is less clear within the chronic-stress stratum. Similarly, among the low-food-security households, chronic stress brings about a statistically significant increase in SSB consumption among children, while the interaction is less evident in the food-secure stratum.

Table 4 presents the effect between household food security and federal aid receipt. It demonstrates that the ORs of low food security are comparable across federal aid levels, indicating that there is no interaction effect between low food security and federal aid on SSB consumption.

Table 5 presents the regression analysis to assess whether parental immigration status modifies the effect of household food security on SSB consumption. It shows that parents who are US born magnify the effect of low food security on increased SSB consumption. Specifically, children in low-food-security households with a US-born parent had higher odds of SSB consumption (OR = 4.2, 95%CI: 2.9–6.3, *p* < 0.001) compared to their counterparts in high-food-security households with immigrant parents. 

## 4. Discussion

Given that food security is a poverty-related condition, this study sought to elucidate the socioeconomic factors that may amplify or offset the negative dietary consequences of low household food security such as greater SSB consumption. Our findings are consistent with the previous literature, which shows that children living in households with low food security had higher SSB consumption [14,15,16,17,18,32]. The novel findings in our study are that (1) parental US nativity strengthens the effect of low food security on SSB consumption among children; (2) chronic stress strengthens the effect of low food security; and (3) there is no interaction between receipt of federal aid and household food security. Specifically, our study suggests that US-born parents in low-food-security households may uphold more permissive SSB consumption norms than immigrant parents. Given evidence that parents determine their children’s SSB consumption [13,14,15,16,17,18,19], this is consistent with past research on nativity and dietary patterns, which suggests that immigrants often consume a more nutritious diet even in the face of a low SES [33,34]. However, this advantage quickly dwindles for ultra-processed foods, [35] and in the case of soda, our study is consistent with research showing SSB consumption increases with acculturation [36]. Longitudinal, in-depth qualitative studies that follow the experience of both US-born and immigrant food-insecure families could shed light on important dietary shifts experienced by children and provide important intervention components, targets, and sequencing for the reducing SSB consumption in this population.

Our findings also suggest the importance of the intersection between food security and stress. Research has shown important differences between acute and chronic stress, with the latter correlating with sustained low-grade inflammation that has severe long-term health consequences [37]. Indeed, Leddy et al. recently proposed a conceptual model whereby food insecurity affects health outcomes through three distinct pathways: (1) inflammation, (2) household stress, and (3) behavioral changes [38]. Although all of these proposed pathways could potentially impact the relationship between low food security and SSB consumption, our research suggests the importance in distinguishing acute vs. chronic forms of stress, and future research should investigate how this might shape the inflammation and behavior of low-food-secure families. Datasets with biometric measures of chronic stress should be mined, given the strong relationship between glucose and stress in animal and human experiments [39,40,41]. That is, glucose may play a role in alleviating cognitive overload and restoring feelings of self-control and may be one of the reasons why those with low food security might turn to sugar, including SSBs. This would add an important dimension to past research that suggests that those experiencing food insecurity consume cheap, highly processed foods to compensate for feeling stressed as well as to obtain more calories for less money [42,43,44].

Finally, we interpret the null findings in light of the gap in access to food amelioration programs here in NYC and the need to incorporate a concept related to food insecurity—that of nutrition security, defined by the USDA as having consistent access, availability, and affordability of foods and beverages that promote well-being and prevent (and if needed, treat) disease [45]. There are no validated measures to assess nutrition security among vulnerable populations in NYC, although measures are being developed in other parts of the country (e.g., Nebraska). Future studies should explore the role of SSBs in preventing achievements in nutrition security as well as how food-specific programs generally or SNAP specifically may help reduce SSB consumption. For example, SNAP Education, a nutrition education component of SNAP, has modules such as “Eat right when money’s tight”, which are designed to lower SSB consumption as well as other non-healthful choices [46].

### Limitations

Our indicator of chronic stress was limited in that it used only one item based on a subjective assessment of the parent/guardian’s ability to cover basic expenses since the child was born; this item might not have captured other essential components of economic security that would be associated with child-level outcomes. Our federal aid indicator pooled all aid from different federally funded sources, which means it sheds little light on specific programs and their impact. The overall response rate for NYC KIDS was only 24.4%, but this is comparable to other telephone-based surveys. For example, the Pew Research Center report indicated a 9% response rate for this type of survey methodology, with a trend of decreasing response rates over time [47]. Our consumption outcomes were not derived using gold standard methodology such as 24 h dietary recalls from the child, as this analysis relied on parental reports of intake. There is a concern of bias, because parents/guardians may underreport the consumption of SSBs, particularly among older children who may consume SSBs in other settings (e.g., friends’ homes). The exclusion of the Latino cultural sweetened beverages is also a limitation. Given the cross-sectional nature of our data, we cannot make any causal statements, and SSBs may be particularly problematic for reverse causation with physical outcomes because some obese children may switch to diet soda as part of a weight-loss strategy [39]. Future research should capture the consumption of diet SSBs if weight outcomes are to be explored.

## 5. Conclusions

Our study found that US parental nativity and chronic stress amplify the association between low household food security and increased SSB consumption among NYC children, which highlights the importance of broader sociodemographic determinants shaping this relationship among vulnerable children in urban settings. 

## Figures and Tables

**Figure 1 nutrients-15-03945-f001:**
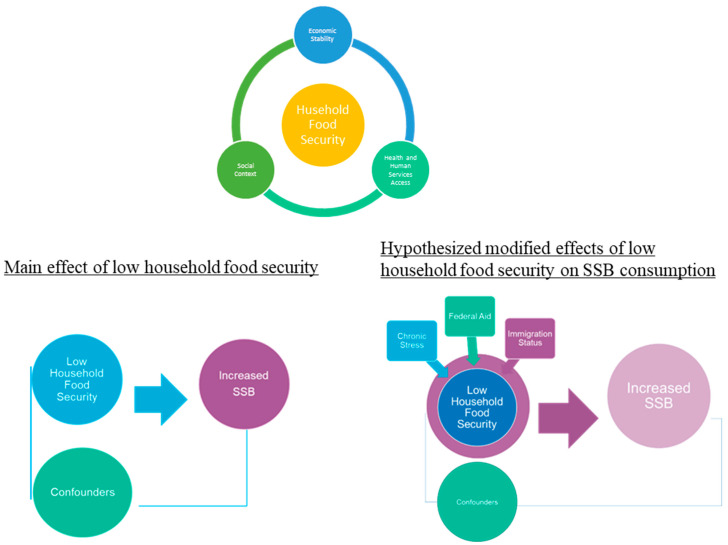
Conceptual figure drawing on a modified Social Determinants of Health Framework; hypothesized relationships.

**Figure 2 nutrients-15-03945-f002:**
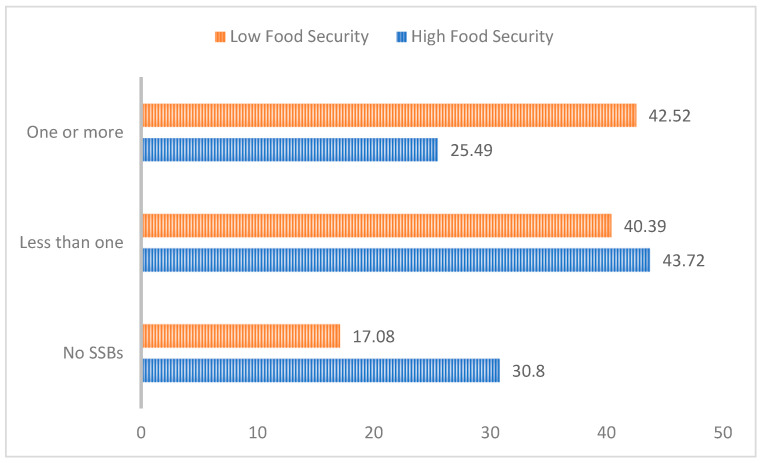
SSB consumption by household food security.

**Table 1 nutrients-15-03945-t001:** Descriptive characteristics by food security levels, NYC Kids 2017.

	Unweighted *n* = 2362 (*n*, %)	High Food Security *n* = 1466 (*n*, %)	Low Food Security *n* = 896 (*n*, %)	*p*-Value
Child’s age, years				
5–13	9.2	9.2 (0.09)	9.2 (0.11)	
Child’s sex				
Male	1236 (50.6)	50.7	50.4	0.909
Female	1125 (49.3)	49.2	49.6	
Child’s race/ethnicity				
White, non-Latino	546 (27.5)	37.2	10.6	<0.001
Black, non-Latino	481 (21.9)	19.0	26.9	
Latino	959 (35.8)	26.9	51.4	
Asian	266 (13.4)	15.6	9.7	
Other, non-Latino	110 (1.2)	1.2	1.3	
Age of Adult Respondent ^1^				
16–24	30 (0.7)	0.82	0.56	<0.001
25–44	1817 (74.7)	71.4	80.6	
45–65+	515 (24.5)	27.8	18.7	
Other Children in Household ^2^				
None	80 (25.2)	26.5	22.9	0.1220
One	952 (40.6)	41.6	38.8	
Two	390 (19.6)	18.7	21.2	
Three	140 (8.2)	6.6	10.9	
More than three	77 (6.4)	6.6	6.0	
Household Poverty				
Federal Poverty Level <200%	1405 (61.5)	49.4	82.8	<0.001
Federal Poverty Level 200%+	957 (38.5)	50.7	17.2	
Chronic Stress				
No	1595 (67.1)	85.1	34.6	<0.001
Yes	767 (32.8)	14.9	65.3	
Receipt of Federal Aid				
No	1576 (65.4)	78.3	42.9	<0.001
Yes	786 (34.5)	21.7	57.1	
Parental Immigration Status ^3^				
Born outside USA	1265 (51.9)	45.5	63.1	<0.001
US born	1097 (48.1)	54.6	36.9	
Parental Education ^3^				
Elementary	186 (7.6)	4.4	13.1	
9–11	219 (9.6)	6.2	15.4	<0.001
Grade 12/GED	588 (27.3)	24.5	32.3	
Some College	467 (18.2)	16.0	22.1	
College+	906 (37.2)	48.7	17.1	

^1^ 88% of respondents were the child’s mother; 12% of respondents were the child’s father. ^2^ Any other children aged 0–13 besides the sampled child. ^3^ Based on main parent being interviewed.

**Table 2 nutrients-15-03945-t002:** Crude and adjusted models for food security on SSB consumption ^a^, NYC Kids 2017.

	Crude				Adjusted ^b^			
		95% CI				95% CI		
	OR	Lower	Upper	*p*-Value	OR	Lower	Upper	*p*-Value
Low Food Security	2.2	1.7	2.7	<0.01	2.0	1.6	2.6	<0.01
High Food Security								

^a^ Soda and other SSBs combined. ^b^ Adjusted for child’s sex (male; female), child’s age (years), child’s race and ethnicity (White; Black; Latino; Asian; Other non-Latino), respondent parent’s age (16–24; 25–44; 45–65), number of other kids in household (none; one; two; three; three or more), household poverty (<200% FPL; 200% + FPL), respondent parent’s education (elementary school; 9–11; high school or GED; college 1–3 years; college 4 years or more). Sample size = 2362.

**Table 3 nutrients-15-03945-t003:** Interaction of food security and chronic stress on odds of sugar-sweetened beverage consumption ^a^, NYC Kids 2017.

	Exposure= 0 (High Food Security) ^b^				Exposure = 1 (Low Food Security) ^b^							
		*n* = 1466				*n* = 896						
	OR	Lower	Upper	*p*-Value	OR	Lower	Upper	*p*-Value	OR of Low Food Security within Strata of Stress			
0 = No stress	Ref				2.3	1.5	3.3	<0.001	2.2	1.5	3.1	<0.001
1 = Stress	1.7	1.2	2.5	<0.05	2.2	1.6	3.0	<0.001	1.4	0.90	2.1	0.128
OR of stress within strata	1.9	1.3	2.7	<0.05	0.96	0.65	1.4	0.845				

^a^ Soda and other SSBs combined. ^b^ Adjusted for child’s sex (male; female), child’s age (years), child’s race and ethnicity (White; Black; Latino; Asian; Other non-Latino), respondent parent’s age (16–24; 25–44; 45–65), number of other kids in household (none; one; two; three; three or more), household poverty (<200% FPL; 200% + FPL), respondent parent’s education (elementary school; 9–11; high school or GED; college 1–3 years; college 4 years or more). Sample size = 2362.

**Table 4 nutrients-15-03945-t004:** Interaction of food security and receipt of federal aid on odds of sugar-sweetened beverage consumption ^a^, NYC Kids 2017.

	Exposure= 0 (High Food Security) ^b^				Exposure = 1 (Low Food Security) ^b^							
		*n* = 1466				*n* = 896						
	OR	Lower	Upper	*p*-Value	OR	Lower	Upper	*p*-Value	OR of Low Food Security within Strata			
0 = No aid	Ref				1.7	1.2	2.3	<0.001	1.7	1.2	3.5	<0.001
1 = Aid	1.1	0.79	1.7	0.410	2.6	1.8	3.8	<0.001	2.3	1.6	3.5	<0.001
OR of aid within strata	1.1	0.76	1.7	0.484	1.6	1.1	2.4	<0.05				

^a^ Soda and other SSBs combined. ^b^ Adjusted for child’s sex (male; female), child’s age (years), child’s race and ethnicity (White; Black; Latino; Asian; Other non-Latino), respondent parent’s age (16–24; 25–44; 45–65), number of other kids in household (none; one; two; three; three or more), household poverty (<200% FPL; 200% + FPL), respondent parent’s education (elementary school; 9–11; high school or GED; college 1–3 years; college 4 years or more). Sample size = 2362.

**Table 5 nutrients-15-03945-t005:** Interaction of food security and immigration on odds of sugar-sweetened beverage consumption ^a^, NYC Kids 2017.

	Exposure= 0 (High Food Security) ^b^				Exposure = 1 (Low Food Security) ^b^							
		*n* = 1466				*n* = 896						
	OR	Lower	Upper	*p*-Value	OR	Lower	Upper	*p*-Value	OR for Low Food Security within Strata			
0 = Immigrant	Ref				1.8	1.4	2.5	<0.001	1.9	1.3	2.5	<0.001
1 = USA	1.6	1.2	2.2	<0.05	4.2	2.9	6.3	<0.001	2.2	1.4	3.4	<0.001

^a^ Soda and other SSBs combined. ^b^ Adjusted for child’s sex (male; female), child’s age (years), child’s race and ethnicity (White; Black; Latino; Asian; Other non-Latino), respondent parent’s age (16–24; 25–44; 45–65), number of other kids in household (none; one; two; three; three or more), household poverty (<200% FPL; 200% + FPL), respondent parent’s education (elementary school; 9–11; high school or GED; college 1–3 years; college 4 years or more). Sample size = 2362.

## Data Availability

The dataset analyzed during the current study is publicly available, though a data use agreement between the authors and the NYC Department of Health and Mental Hygiene was obtained for the present study.

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
