# Peer review of "Household Food Security and Consumption of Sugar-Sweetened Beverages among New York City (NYC) Children: A Cross-Sectional Analysis of 2017 NYC Kids’ Data"

_nutrients, 2023, doi:10.3390/nu15183945_

Round 1
Reviewer 1 Report
please show the SDOH framework and demonstrate how it was used since you say , "we draw on the more expansive social determinants of health (SDOH) framework to examine how other important socioeconomic factors may serve to...."
I suggest the authors more clearly elucidate the missing factors that could explain their outcomes in a table. The authors might explain how qualitative data could help to elucidate the factors that explain non native consumption of SSB. Last in figure 1 the content does not march the figure. The authors use these terms which are not in the diagram:" ..... we expected this relationship to be magnified by additional vulnerabilities such as chronic hardship, receipt of federal aid, and parental nativity". Then figure 1 shows confounding factors but missed or did not include a discussion of confounding factors in the content.
Author Response
Thank you for your review of our paper. We are grateful for the input and the opportunity to address any concerns. As a result, our paper has been significantly strengthened. Below please find our response (in regular type) to each reviewer comment (in bold). Changes within the manuscript are tracked.
Reviewer 1
Please show the SDOH framework and demonstrate how it was used since you say , "we draw on the more expansive social determinants of health (SDOH) framework to examine how other important socioeconomic factors may serve to...."
We now explicitly state on page 3 how we drew from the Healthy People 2030 framework of Social Determinants of Health, as well as integrated these components in Figure 1.
I suggest the authors more clearly elucidate the missing factors that could explain their outcomes in a table.
The confounding variables, along with the variables of interest, are listed in Table 1.
The authors might explain how qualitative data could help to elucidate the factors that explain nonnative consumption of SSB.
Thank you we now include this on page 10 of the discussion section.
Last in figure 1 the content does not march the figure. The authors use these terms which are not in the diagram:" ..... we expected this relationship to be magnified by additional vulnerabilities such as chronic hardship, receipt of federal aid, and parental nativity".
Thank you we now use consistent language between the figure and text throughout the manuscript.
Then figure 1 shows confounding factors but missed or did not include a discussion of confounding factors in the content.
Thank you we now use consistent language but we do not feel the need to discuss confounding factors in the discussion since they are not our main effect variables. However, we have relabeled this section in the methods section to match the figure.

Reviewer 2 Report
The text of the work has not been carefully prepared, e.g. references have not been arranged according to the order of citation in the text and not all abbreviations have been explained, e.g. pre-K (line 162)
The introduction should be refined, selected fragments should be ordered and transferred to the discussion, this part of the work lacks a clearly defined purpose of the work.
In the Data Source section, the note in line 105-107 should be included in the discussion. Also there is no explanation in the text of the work about household poverty level (<200%, 200%+) and its meaning (Line 157).
In the Analysis part, there is no information about the qualification criteria of the respondents for the following groups: a) high food security/no stress (reference) b) high food security/stress c) low food security/no stress d) low food security/stress?
In the Results section In table 1 there is a lack of unification of the presented data and there are missing numerical data. For example, the division of parents' education used in the assessment (elementary school; 9-11; High school or GED; College 1-3 years; College 4+) is presented in Table 1 as (only college; college +)
Figure 2 does not show numerical data.
Table 3 - we do not know on what basis the occurrence of stress and its absence were determined.
The discussion is rudimentary and not properly developed.
The conclusions needs to be shortened.
Author Response
Thank you for your review of our paper. We are grateful for the input and the opportunity to address any concerns. As a result, our paper has been significantly strengthened. Below please find our response (in regular type) to each reviewer comment (in bold). Changes within the manuscript are tracked.
Reviewer 2
The text of the work has not been carefully prepared, e.g. references have not been arranged according to the order of citation in the text and not all abbreviations have been explained, e.g. pre-K (line 162)
All references have been fixed. We also spelled out pre-K on line 162 as well SNAP, WIC , GED and all other acronyms in the paper. Please note that once we have spelled out the term, we only use the acronym thereafter.
The introduction should be refined, selected fragments should be ordered and transferred to the discussion, this part of the work lacks a clearly defined purpose of the work.
Thank the introduction has refined per your suggestion and we have added the following sentence in the Introduction section to better convey the purpose of our paper:
“This paper investigates whether, and if so, how, socioeconomic factors (receipt of federal aid, parental immigration status, and chronic stress) modify the association between low food security of households and SSB consumption among children.”
In the Data Source section, the note in line 105-107 should be included in the discussion. Also, there is no explanation in the text of the work about household poverty level (<200%, 200%+) and its meaning (Line 157).
We have moved the note in lines 105-107 about typical response rates reported by Pew Research Center to the Discussion section. We have added the note and now explain household poverty level measure more clearly.
In the Analysis part, there is no information about the qualification criteria of the respondents for the following groups: a) high food security/no stress (reference) b) high food security/stress c) low food security/no stress d) low food security/stress?
The inclusion criteria for food security and chronic stress are described in Section 2: Materials and Methods under “Exposures”. We made this clearer by noting that the paragraph for “Chronic stress” in Section 2 relates to the stress/no stress qualification criteria.
In the Results section In table 1 there is a lack of unification of the presented data and there are missing numerical data. For example, the division of parents' education used in the assessment (elementary school; 9-11; High school or GED; College 1-3 years; College 4+) is presented in Table 1 as (only college; college +)
Thank you we have fixed this issue and all variables as described in the methods text match Table 1.
Figure 2 does not show numerical data.
Figure 2 now shows percentages.
Table 3 - we do not know on what basis the occurrence of stress and its absence were determined.
Thank you, please look at page 5 for the specific questions that were asked to determine stress as stated in the "Chronic hardship" paragraph in Section 2: Materials and Methods.
The discussion is rudimentary and not properly developed.
We have added more discussion regarding our results and reflect to the extent to which our findings are in line with previous literature.
The conclusions need to be shortened.
We have shortened the conclusion to mention just three suggestions for future research.
